# Metabolomic adaptations and correlates of survival to immune checkpoint blockade

Haoxin Li[1,2,3], Kevin Bullock[2], Carino Gurjao[1,2], David Braun[1], Sachet A. Shukla[1], Dominick Bossé [1], Aly-Khan A. Lalani [1], Shuba Gopal[2], Chelsea Jin[4], Christine Horak[4], Megan Wind-Rotolo[4], Sabina Signoretti[1], David F. McDermott[5], Gordon J. Freeman [1], Eliezer M. Van Allen [1,2,6], Stuart L. Schreiber[2,3], F. Stephen Hodi[1,6], William R. Sellers [1,2], Levi A. Garraway[1,2,6], Clary B. Clish [2], Toni K. Choueiri [1,6,7]* & Marios Giannakis[1,2,6,7]*

Despite remarkable success of immune checkpoint inhibitors, the majority of cancer patients have yet to receive durable benefits. Here, in order to investigate the metabolic alterations in response to immune checkpoint blockade, we comprehensively profile serum metabolites in advanced melanoma and renal cell carcinoma patients treated with nivolumab, an antibody against programmed cell death protein 1 (PD1). We identify serum kynurenine/tryptophan ratio increases as an adaptive resistance mechanism associated with worse overall survival. This advocates for patient stratification and metabolic monitoring in immunotherapy clinical trials including those combining PD1 blockade with indoleamine 2,3-dioxygenase/tryptophan 2,3-dioxygenase   (IDO/TDO) inhibitors.

[1] Department of Medical Oncology, Dana-Farber Cancer Institute, Boston, MA 02215, USA. [2] Broad Institute of MIT and Harvard, Cambridge, MA 02142, USA. [3] Department of Chemistry and Chemical Biology, Harvard University, Cambridge, MA 02138, USA. [4] Bristol-Myers Squibb, Princeton, NJ 08540, USA. [5] Beth Israel Deaconess Medical Center, Boston, MA 02215, USA. [6] Department of Medicine, Brigham and Women's Hospital, Harvard Medical School, Boston, MA 02115, USA. [7] These authors jointly supervised this work: Toni K. Choueiri, Marios Giannakis. *email: Toni_Choueiri@dfci.harvard.edu; Marios_Giannakis@dfci.harvard.edu

nhibition of immune-checkpoint targets including PD1 is clinically effective in a variety of cancers[1–4]. However, only a subset of patients respond and complete response remains uncommon. To understand the mechanisms of response and resistance, recent studies have focused on neoantigens[5,6], copy-number alterations[7], and transcriptional signatures[8,9] of tumor tissues collected from patients treated with immune-checkpoint inhibitors. Given the known role of metabolites in modulating immunity[10], we sought to understand how individual patients' metabolic activities adapt to PD1 immune checkpoint blockade and how they associate with therapeutic benefits.

In the present study, by comprehensively profiling serum samples, we find prevalent kynurenine increases in both melanoma and renal cell carcinoma (RCC) patients after receiving nivolumab. Additionally, increased kynurenine metabolism correlates with worse overall survival. Our findings have implications for the design and interpretation of novel combination therapies involving checkpoint inhibition.

## Results

**Metabolomic profiling of serum samples.** We profiled 106–202 metabolites in pre- and multiple on-treatment patient serum samples from three independent immunotherapy trials using liquid chromatography-mass spectrometry (LC-MS) (Fig. 1, Supplementary Data 1, 2 and 3). These metabolites are involved in the metabolism of amino acids, nucleotides, nitrogen, and lipids, among others. Our study consisted of two Phase I trials (CA209-038, NCT01621490; CA209-009, NCT01358721), which included 78 patients with advanced melanoma and 91 patients with metastatic RCC treated with nivolumab[8,11]. To investigate the generalizability of our results, we also analyzed a large randomized Phase III trial (CheckMate 025, NCT01668784) with 743 RCC patients, among which 394 received nivolumab and 349 received everolimus[2]. The basic clinical and demographic features of these patients are described in Supplementary Tables 1, 2 and 3.

**Increases of kynurenine among patients receiving nivolumab.** We first investigated the impact of nivolumab on serum metabolites during treatment compared to the baseline (prior to treatment). Among melanoma patients, kynurenine, a product of

tryptophan catabolism, was the most significantly changed metabolite at week 4 and at week 6 compared to pre-treatment levels (37% and 34% increase on average, $q < 1 \times 10^{-10}$ and $q < 1 \times 10^{-8}$, respectively, paired $t$-test) (Fig. 2a, Supplementary Fig 1a). By using Kyn/Trp as a metric indicating tryptophan-kynurenine conversion, we found that this ratio falls in a range spanning approximately eight-fold among these patients, suggesting prominent individual-to-individual differences (Fig. 2b). Specifically, 78% patients had any increases and 26.5% patients had increases above 50% at week 4. On average, we did not observe further kynurenine increases between week 4 and 6 (Fig. 2c). For RCC patients treated with nivolumab in either the phase 1 or the phase 3 trials, we confirmed kynurenine as a top up-regulated metabolite following nivolumab treatment (Fig. 2d, Supplementary Fig. 1b–d). In particular, the phase 3 cohort showed 23% and 24% increase on average at week 4 and week 8, respectively, ($q < 1 \times 10^{-10}$ and $q < 1 \times 10^{-12}$, paired $t$-test). Additionally, 69.4% and 8.2% patients had Kyn/Trp increases above zero and 50% respectively at week 4 (Fig. 2e). We did not find further increases of kynurenine after week 4 (Supplementary Fig. 1e, f). Notably, patients receiving everolimus control treatment had a decrease in kynurenine (21% decrease on average, $q < 1 \times 10^{-5}$, paired $t$-test) (Fig. 2f) and 69% patients had reduced Kyn/Trp ratios.

Kynurenine is synthesized during tryptophan catabolism by indoleamine 2,3-dioxygenase (IDO) or tryptophan 2,3-dioxygenase (TDO), and has been shown to suppress anti-tumor immune responses[12,13]. To explore whether the circulating Kyn/Trp correlates with immune-suppression in the tumor microenvironment, we analyzed RNAseq data of tumor biopsies from the CA209-038 trial[14]. First, among all tumors sequenced, we observed on average increased transcription of PD-L1 after nivolumab treatment among other immune-suppression regulators including PD1, CTLA4, LAG3, TIM3, and FOXP3 (Supplementary Fig. 2a, b), suggesting up-regulation of an immune-resistance cell program at week 4. Second, we found a significant correlation between Kyn/Trp ratio and PD-L1 expression 4 weeks after starting nivolumab treatment (Pearson correlation, $p = 0.01$; see Fig. 2g). In contrast, Kyn/Trp was not associated with prior anti-CTLA4 treatment, or tumor mutational load (Fig. 2g). Thus, the overall increase in kynurenine levels observed among patients

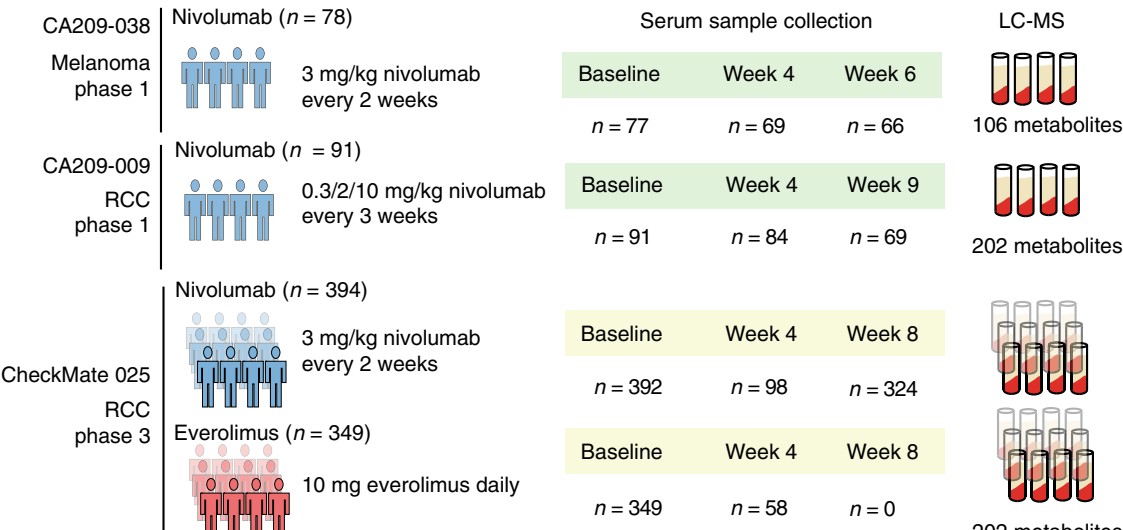

**Fig. 1** Schematic of study design and serum specimen collection. This study consisted of serum specimens from two phase 1 trials and one randomized phase 3 trial. The cancer and treatment type/dosing, number of serum samples collected at each time point, and number of metabolites identified by LC-MS are labeled as above. 63 overlapping metabolites were profiled in all three cohorts

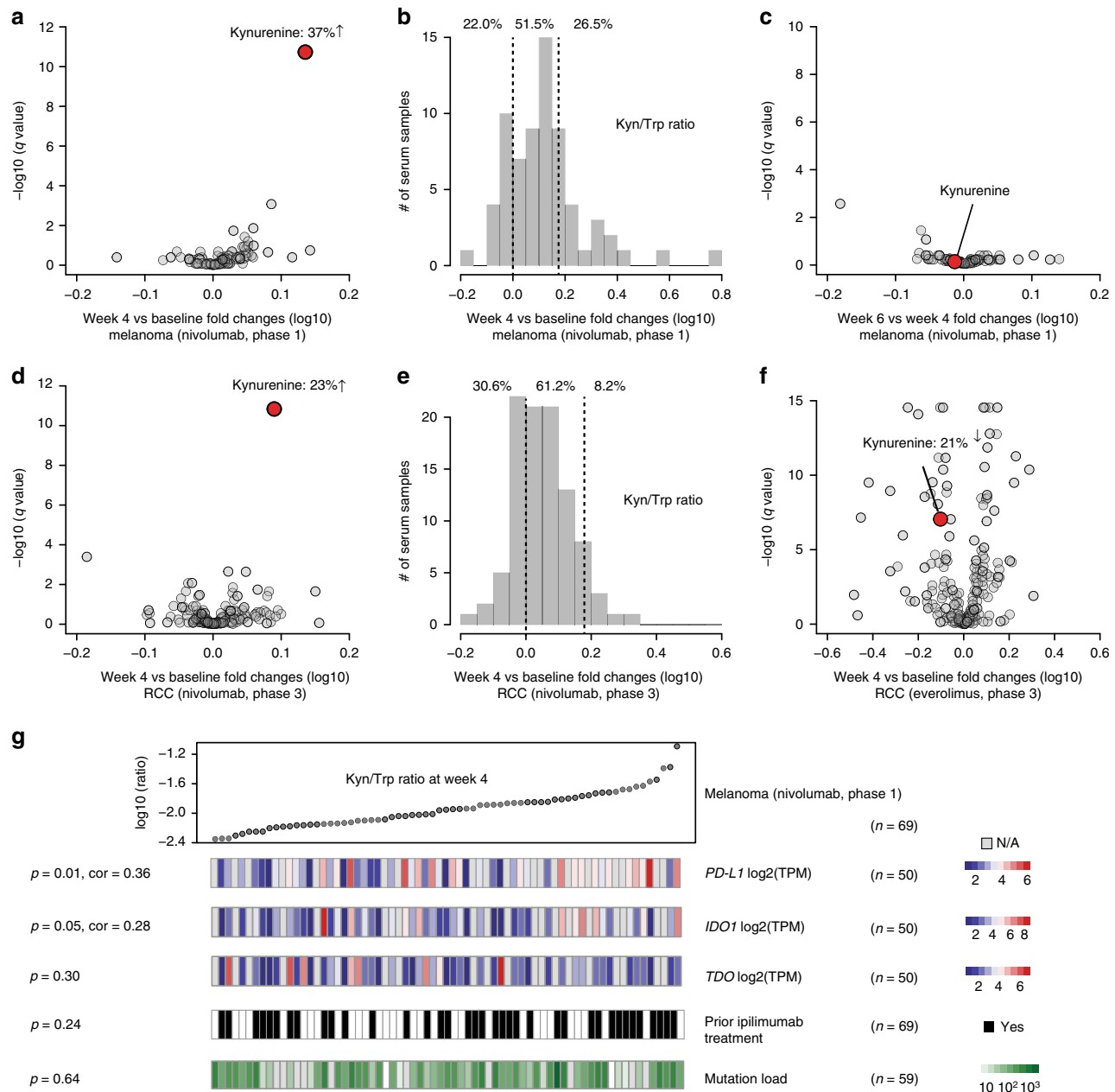

**Fig. 2** Comprehensive serum metabolomic profiling reveals significantly up-regulated kynurenine in response to nivolumab treatment. **a** Volcano plots showing average serum metabolite (n = 106, represented as points) level changes after 4 weeks of nivolumab treatment compared to baseline in CA209-038 melanoma patients. **b** Histogram showing the changes of kynurenine/tryptophan (Kyn/Trp) ratios 4 weeks after nivolumab treatment compared to baseline in melanoma patients. **c** Volcano plot showing average serum metabolite (n = 106) changes between week 6 and week 4 after nivolumab treatment in melanoma patients. **d** Volcano plots showing average serum metabolite (n = 202) level changes after 4 weeks of nivolumab treatment compared to baseline in CheckMate 025 RCC patients. **e** Histogram showing the changes of Kyn/Trp ratios 4 weeks after nivolumab treatment compared to baseline in CheckMate 025 RCC patients. **f** Volcano plot showing average serum metabolite (n = 202) changes after 4 weeks of everolimus treatment compared to baseline in CheckMate 025 RCC patients. **g** Pearson correlation analysis between Kyn/Trp ratios and PD-L1/IDO1/TDO mRNA expression at week 4, prior anti-CTLA4 treatment (ipilimumab), and tumor mutation load in melanoma patients. The q values in **a**, **c**, **d**, **f** were calculated based on paired t-tests for all profiled metabolites with Benjamini-Hochberg multiple testing corrections

in response to nivolumab (but not everolimus) treatment might result in an adaptive immune suppressive microenvironment counterbalancing checkpoint blockade and contribute to immunotherapy resistance independent of previously described genomic correlates[5,14].

**Higher Kyn/Trp ratios associate with worse overall survival.** We then sought to determine if the degrees of serum Kyn/Trp

alterations correlate with clinical outcomes, focusing on overall survival (OS) for patients treated with nivolumab. Compared to all other metabolites, increases of the Kyn/Trp ratio in the melanoma cohort (week 4 or 6 vs baseline) were consistently associated with greater risks for death ($p = 1.2 \times 10^{-4}$, HR = 2.71, 95% CI, 1.63–4.51; $p = 2.5 \times 10^{-4}$, HR = 2.26, 95% CI, 1.46–3.50, respectively, Cox proportional hazards model) (Fig. 3a, b). In contrast, the baseline Kyn/Trp ratio did not

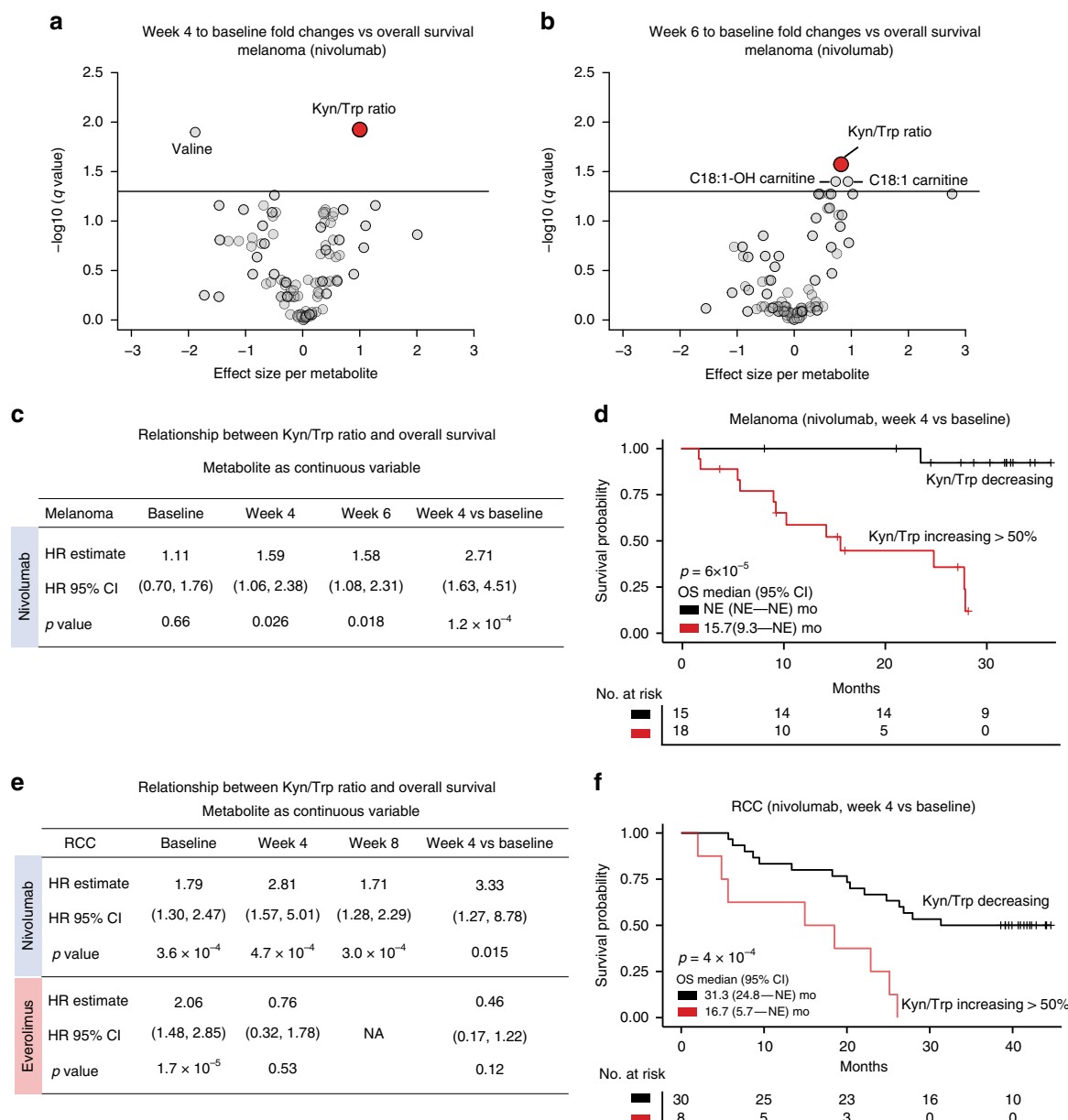

**Fig. 3** Kyn/Trp ratio alterations associate with patient overall survival in two independent cohorts. **a**, **b** Volcano plots showing associations between fold changes (log2 scale) of different serum metabolites ($n = 106$) and overall survival in melanoma patients. The effect sizes refer to the regression coefficients in a Cox proportional hazards model and the points represent different metabolites. **a** 4 weeks after nivolumab treatment versus baseline, **b** 6 weeks after nivolumab treatment versus baseline. $q$ values were calculated using Benjamini-Hochberg multiple testing corrections. A cutoff at $q = 0.05$ is shown as a horizontal line. **c** Table summarizing the hazard ratios (HR) of Kyn/Trp (log2 scale) as a predictor at different time points in relation to melanoma patient overall survival using a Cox proportional hazards model. CI, confidence interval. **d** Kaplan–Meier plot comparing the overall survival in melanoma patients with >50% increases in Kyn/Trp ratios versus those with decreases. **e**, Table summarizing the hazard ratios (HR) of Kyn/Trp (log2 scale) as a predictor at different time points in relation to CheckMate 025 RCC patient overall survival using a Cox proportional hazards model. **f** Kaplan–Meier plot comparing the overall survival in RCC patients with >50% increases in Kyn/Trp ratios versus those with decreases. The $p$ values in **d**, **f** were based on log-rank tests

significantly associate with the melanoma patients' overall survival ($p = 0.66$, HR = 1.11, 95% CI, 0.70–1.76, Cox model) but became significant at week 4 and week 6 (Fig. 3c). This was the case even after adjusting for patient demographic characteristics and *BRAF* mutational status (Supplementary Table 4). In particular, patients with a >50% increase in Kyn/Trp had a median OS of 15.7 months while those with decreases had a median survival time of >38 months (Fig. 3d) ($p = 6.0 \times 10^{-5}$, log-rank test).

To confirm this result, the association between Kyn/Trp ratios and OS in a larger phase 3 trial (CheckMate 025) was evaluated using serum samples collected at different time points. We found that at baseline, higher Kyn/Trp ratios associated with shorter overall survival both for the nivolumab- and the everolimus-treated patients ($p = 3.6 \times 10^{-4}$, HR = 1.79, 95% CI, 1.30–2.47; $p = 1.7 \times 10^{-5}$, HR = 2.06, 95% CI, 1.48–2.85; Cox model) (Fig. 3e, Supplementary Fig. 3a, b). However, at week 4, Kyn/Trp significantly associated with overall survival only in the nivolumab

arm ($p = 4.7 \times 10^{-4}$, HR = 2.81, 95% CI, 1.57–5.01; Cox model) but not in the everolimus arm ($p = 0.53$, HR = 0.76, 95% CI, 0.32–1.78; Cox model) (Fig. 3e, Supplementary Fig. 3c, d). After adjusting for patient demographic characteristics, prior anti-angiogenic treatments, and MSKCC risk group, similar results were obtained (Supplementary Table 5). For nivolumab-treated RCC patients, those with a >50% increases of Kyn/Trp had a median survival of 16.7 months while those with any Kyn/Trp decreases had a median survival of 31.3 months ($p = 4.3 \times 10^{-4}$, log-rank test) (Fig. 3f).

In examining the relationship between baseline Kyn/Trp and MSKCC risk scores (established clinical prognostic factors in advanced RCC)[15], we found that patients with intermediate or poor prognosis had significantly higher baseline levels of Kyn/Trp both in the nivolumab arm (18% and 24% higher mean levels compared to the favorable group, $p = 5.2 \times 10^{-9}$ and $5.5 \times 10^{-9}$, respectively, linear regression) and in the everolimus arm (10% and 23% higher compared to the favorable group, $p = 2.3 \times 10^{-3}$ and $7.0 \times 10^{-7}$ respectively, linear regression) (Supplementary Table 6). However, the change of Kyn/Trp between week 4 and baseline was not associated with MSKCC risk scores in either treatment arm ($p > 0.1$, linear regression) (Supplementary Table 6). Taken together, these results suggest that while baseline Kyn/Trp is associated with disease status prior to treatment and is a prognostic marker for RCC patient survival, its dynamic alteration is independent of pre-treatment disease status.

## Discussion

In summary, we have identified increased tryptophan to kynurenine conversion in response to PD1 blockade in a subset of melanoma and RCC patients. By using two independent cohorts, we showed that Kyn/Trp temporal alterations robustly correlated with overall survival of patients receiving nivolumab. The sources that contributed to these kynurenine changes in serum are unclear. We previously profiled the kynurenine levels in 928 cancer cell lines and found marked outlier production of kynurenine and that secreted kynurenine can be attributed to *IDO* or/and *TDO* expression[16]. These data suggest that tumor cells could be a source of the kynurenine response. Here, we found a correlation between Kyn/ Trp and *IDO1* but not *TDO* mRNA levels in melanoma samples 4 weeks after nivolumab treatment (Fig. 2g). However, besides tumor, other sources of host-derived tryptophan to kynurenine conversion (e.g., macrophages[17], dendritic cells[18,19]) cannot be ruled out.

Earlier studies demonstrated that increased tryptophan to kynurenine conversion leads to inhibition of T cell proliferation[17–19]. By suppressing this pathway, tumor immune resistance could be reversed[12,13] and checkpoint inhibition efficacy could be enhanced in animal models[20]. Our findings further illustrate that checkpoint blockade in combination with IDO/TDO inhibitors might only benefit a selected group of patients with checkpoint-inhibition-triggered kynurenine pathway activation. Given the lack of improved therapeutic outcomes with PD1 and selective IDO1 inhibition among unselected patient populations in the recent phase 3 ECHO-301/KEYNOTE-252 trial[21], our findings highlight the need and feasibility of patient stratification by monitoring serum Kyn/Trp alterations and more generally point to the relevance of metabolic adaptations in cancer immunotherapy. Moreover, kynurenine production or kynurenine signaling may still be a relevant therapeutic target.

## Methods

**Patient population**. Study design, eligibility criteria, and treatment were previously described for Bristol-Myers Squibb trials CA209-038 (NCT01621490)[11], CA209-009 (NCT01358721)[8], and CheckMate 025 (NCT01668784)[2] that enrolled patients

with histologically confirmed diagnoses of advanced melanoma or metastatic RCC. The patients provided written informed consent and the study protocol was approved by the Institutional Review Board of the Dana-Farber Cancer Institute/ Dana-Farber/Harvard Cancer Center. The demographic and clinical characteristics of patients participated in this study are included in Supplementary Tables 1–3.

**Serum sample collection and processing**. Serum was collected at the specified time-points by centrifugation at 4000 g for 4 min at 25 °C within 2 h of collection. Samples were frozen immediately and stored at or below −20 °C for up to 2 months followed by storage at −80 °C. The metabolites were profiled using liquid chromatography-mass spectrometry (LC-MS).

**Metabolomic profiling of Serum Samples from CA209-038**. Positive ionization mode data were acquired using a 6495 triple quadrupole mass spectrometer coupled to a 1290 Infinity II U-HPLC system (Agilent, Santa Clara, CA). Serum samples (10 μL) were extracted using 90 μL of 74.9:24.9:0.2 (v/v/v) acetonitrile/ methanol/formic acid containing stable isotope-labeled internal standards (0.2 ng/μL valine-d8, Isotec; 0.2 ng/μL phenylalanine-d8, Cambridge Isotope Laboratories). The samples were centrifuged (10 min, 9000 g, 4 °C) and the supernatants (10 μL) were injected onto a 150 × 2.1 mm Atlantis HILIC column (Waters). The column was eluted isocratically at a flow rate of 250 μL/min with 5% mobile phase A (10 mM ammonium formate and 0.1% formic acid in water) for 0.5 min followed by a linear gradient to 40% mobile phase B (acetonitrile with 0.1% formic acid) over 10 min. MS data were acquired using multiple reaction monitoring. Retention times, mass transitions, and collision energies were determined using authentic reference standards. Other MS parameters were: ion spray voltage, 3.0 kV; source temperature, 200 °C; nozzle voltage, 500 V; gas flow, 14 L/min; nebulizer, 40 psi; sheath gas, 250 °C; sheath gas flow, 1 L/min; iFunnel high pressure RF, 90; and low pressure RF, 90. Raw data were processed using MassHunter software (Agilent, Santa Clara, CA) for automated peak integration. The peak areas of 106 targeted metabolites were manually reviewed for quality of integration and compared against standard reference standards to confirm identities.

**Metabolomic profiling of Serum Samples from CA209-009 and Check-Mate025**. High resolution, accurate mass LC-MS data were acquired using identical sample preparation and chromatography conditions, but using a system comprised of a Shimadzu Nexera X2 U-HPLC (Shimadzu Corp.; Marlborough, MA) coupled to a Q Exactive hybrid quadrupole orbitrap mass spectrometer (Thermo Fisher Scientific; Waltham, MA). MS analyses were carried out using electrospray ionization in the positive ion mode using full scan analysis over 70–800 m/z at 70,000 resolution and 3 Hz data acquisition rate. Other MS settings were: sheath gas 40, sweep gas 2, spray voltage 3.5 kV, capillary temperature 350 °C, S-lens RF 40, heater temperature 300 °C, microscans 1, automatic gain control target 1e6, and maximum ion time 250 ms. Raw data were processed using TraceFinder software (Thermo Fisher Scientific; Waltham, MA) and Progenesis QI (Nonlinear Dynamics; Newcastle upon Tyne, UK). The identities of 202 profiled metabolites were confirmed using reference standards.

**Metabolomic data processing and quantification**. Internal standards including valine-d8 and phenylalanine-d8 were used to validate consistent sample processing. In each dataset, pooled plasma reference samples were included in the analytical queue at intervals of 20 samples. To mitigate risk of temporal drift in instrument response over the course of the analyses, data for each peak (metabolite) were standardized using the ratio between the value in each sample and the nearest pooled reference multiplied by the median peak area among all pooled references. Absolute quantitation of plasma tryptophan and kynurenine was performed using external calibration curves prepared via serial dilution of stable isotope-labeled reference standards including L-tryptophan ($^{13}$C11, 99%, Cambridge Isotope Laboratories) and L-kynurenine (D6, 97% +, Cambridge Isotope Laboratories) in human plasma.

**Statistical analysis**. All statistical analyses were done in R v 3.2.1 (downloaded from https://www.r-project.org/). All statistical tests were two-sided. Statistics and relevant information including the adopted statistical tests and p-values are reported in the figures and associated legends. Benjamini-Hochberg multiple testing corrections were applied to obtain q values. For Pearson correlations, we used the *cor.test* function in R to conduct significance tests and obtain the p-values (two-sided). For survival analysis, we used the *survival* package in R.

**Reporting summary**. Further information on research design is available in the Nature Research Reporting Summary linked to this article.

## Data availability

The metabolomic data and patient information supporting the findings of this study are available in the Supplementary Data files. The raw LC-MS source data generated in this study have been deposited at www.metabolomicsworkbench.org with the project ID PR000828. The RNASeq data[14] were downloaded from (SRA: SRP094781; BioProject: PRJNA356761).

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

## Acknowledgements

We thank the patients and their families as well as faculty and personnel of investigating sites. We also would like to thank Diana Miao, Claire Margolis and Jihye Park as well as Kaushal Desai, Michael Manos, and Mariano Severgnini for help with various aspects of this research. M.G. is supported with a Conquer Cancer Foundation ASCO Career Development Award (mentor: T.K.C.), and a Stand Up To Cancer Colorectal Cancer Dream Team Translational Research Grant (grant number: SU2C-AACR-DT22-17). Stand Up To Cancer is a division of the Entertainment Industry Foundation. Research grants are administered by the American Association for Cancer Research, the Scientific Partner of SU2C. This research was also supported in part by the Dana-Farber/Harvard Cancer Center Kidney Cancer program (D.F.M, T.K.C., S.S.), the NCI's Cancer Target Discovery and Development (CTD2) Network (grant number U01CA217848, awarded to S.L.S.), Kidney SPORE P50CA101942-12 (D.F.M., G.J.F, T.K.C., S.S.), and the Trust Family, Michael Brigham, and Loker Pinard Funds for Kidney Cancer Research at Dana-Farber Cancer Institute (T.K.C.). This research was also supported by the BMS II-ON consortium (T.K.C., M.G., F.S.H.) and the AACR KureIt Grant for Kidney Cancer (E.M.V., T.K.C.).

## Author contributions

H.L., L.A.G., T.K.C. and M.G. designed research. H.L., K.B., C.G., D.Br., S.A.S., D. Bo., A.A.L., S.G., C.J. and M.G. performed research. C.H., M.W., S.S., D.F.M, G.J.F., E.M.V., S.L.S., F.S.H., W.R.S., L.A.G., C.B.C., T.K.C. and M.G. contributed reagents and resources. L.H., T.K.C. and M.G. wrote the paper in discussion with other co-authors.

## Competing interests

C.J., C.H. and M.W. are employees of Bristol-Myers Squibb. G.J.F. has patents/pending royalties on the PD-1 pathway from Roche, Merck, Bristol Myers-Squibb, EMD-Serono, Boehringer-Ingelheim, AstraZeneca, Dako, and Novartis and has served on advisory boards for CoStim, Novartis, Roche, Eli Lilly, Bristol-Myers-Squibb, Seattle Genetics, Bethyl Laboratories, Xios, and Quiet. F.S.H. receives consulting fees from Bristol Myers-Squibb, Merck, EMD-Serono, Novartis, Celldex, Amgen, Genentech/Roche, Incyte, Bayer, Partners Therapeutics, Sanofi, Pfizer and is on the advisory board for Apricity, Aduro, Pionyr, 7 Hills Pharma, Verastem, Compass Therapeutics, Takeda and holds equity and is on the advisory board for Torque. L.A.G. was a paid consultant for Novartis, Foundation Medicine, and Boehringer-Ingelheim; he held equity in Foundation Medicine and was a recipient of a grant from Novartis. L.A.G. is an employee of Roche. M.G. receives research funding from Bristol Myers-Squibb and Merck. T.K.C. receives research funds from AstraZeneca, Bayer, BMS, Cerulean, Eisai, Foundation Medicine Inc., Exelixis, Ipsen, Tracon, Genentech, Roche, Roche Products Limited, GlaxoSmithKline, Merck, Novartis, Peloton, Pfizer, Prometheus Labs, Corvus, Calithera, Analysis Group, Takeda. T.K.C. receives honoraria from AstraZeneca, Alexion, Sanofi/Aventis, Bayer, BMS, Cerulean, Eisai, Foundation Medicine Inc., Exelixis, Genentech, Roche, GlaxoSmithKline, Merck, Novartis, Peloton, Pfizer, EMD Serono, Prometheus Labs, Corvus, Ipsen, Up-to-Date, Analysis Group, NCCN, Michael J. Hennessy (MJH) Associates, Inc (Healthcare Communications Company with several brands such as OnClive and PER), L-path, Kidney Cancer Journal, Clinical Care Options, Platform Q, Navinata Healthcare, Harborside Press, American Society of Medical Oncology, NEJM, Lancet Oncology, Heron Therapeutics. T.K.C has consulting or advisory role for AstraZeneca, Alexion, Sanofi/Aventis, Bayer, BMS, Cerulean, Eisai, Foundation Medicine Inc., Exelixis, Genentech, Heron Therapeutics, Roche, GlaxoSmithKline, Merck, Novartis, Peloton, Pfizer, EMD Serono, Prometheus Labs, Corvus, Ipsen, Up-to-Date, NCCN, Analysis Group. T.K.C. does not serve within a speaker's bureau. T.K.C. does not have leadership or employment in for-profit companies. Other present or past leadership roles of T.K.C. include Director of GU Oncology Division at Dana-Farber and past President of medical Staff at Dana-Farber), member of NCCN Kidney panel and the GU Steering Committee, past chairman of the Kidney cancer Association Medical and Scientific Steering Committee). H.L., T.K.C. and M.G. have pending patents for biomarkers of immune checkpoint blockers. T.K.C. has stock ownership in Pionyr and Tempest. For T.K.C., travel, accommodations, and expenses, in relation to consulting, advisory roles, honoraria, medical writing and editorial assistance support may have been funded by communications companies funded by pharmaceutical companies. The institution (Dana-Farber Cancer Institute) may have received additional independent funding of drug companies or/and royalties potentially involved in research around the subject matter. The remaining authors declare no competing interests.
