## [Peer Review File · Nature Communications]

REVIEWERS' COMMENTS:

Reviewer #2 (Remarks to the Author):

This manuscript has been significantly improved since initial filing.

The data provided in the Excel Data files now fully support the message illustrated in the figures about the increased Kyn/Trp ratio in patients who do not respond to anti-PD1 (first revised version).

In this second revised version, as requested in my comments, the authors have added IDO/TDO expression data in the tumor samples of the same cohort of patients, in an effort to understand the origin of this increased Kyn/Trp ratio. In melanoma, they observed a correlation between Kyn/Trp ratios and tumoral expression of IDO but not TDO. Although at the limit of significance, this correlation makes sense and supports the notion that IDO can be induced, along with other immunosuppressive factors, in some melanoma patients after anti-PD1 therapy, and thereby contribute to therapy resistance.

The findings reported in this manuscript highlight the interest of metabolic profiling and provide a means to stratify patients and better define the place of IDO inhibitors in cancer immunotherapy.

Reviewer #4 (Remarks to the Author):

The authors have done a good job overall in addressing the previous concerns.

REVIEWERS' COMMENTS:

Reviewer #2 (Remarks to the Author):

This manuscript has been significantly improved since initial filing.

The data provided in the Excel Data files now fully support the message illustrated in the figures about the increased Kyn/Trp ratio in patients who do not respond to anti-PD1 (first revised version).

In this second revised version, as requested in my comments, the authors have added IDO/TDO expression data in the tumor samples of the same cohort of patients, in an effort to understand the origin of this increased Kyn/Trp ratio. In melanoma, they observed a correlation between Kyn/Trp ratios and tumoral expression of IDO but not TDO. Although at the limit of significance, this correlation makes sense and supports the notion that IDO can be induced, along with other immunosuppressive factors, in some melanoma patients after anti-PD1 therapy, and thereby contribute to therapy resistance.

The findings reported in this manuscript highlight the interest of metabolic profiling and provide a means to stratify patients and better define the place of IDO inhibitors in cancer immunotherapy.

Reviewer #4 (Remarks to the Author):

The authors have done a good job overall in addressing the previous concerns.

Reply to referee comments:

Response: We thank all the reviewers for their feedback and we are glad that they are satisfied with our revisions.